# The Advancement of Biodegradable Polyesters as Delivery Systems for Camptothecin and Its Analogues—A Status Report

**DOI:** 10.3390/ijms24021053

**Published:** 2023-01-05

**Authors:** Katarzyna Strzelecka, Urszula Piotrowska, Marcin Sobczak, Ewa Oledzka

**Affiliations:** 1Department of Analytical Chemistry and Biomaterials, Faculty of Pharmacy, Medical University of Warsaw, 1 Banacha Str., 02-097 Warsaw, Poland; 2Military Institute of Hygiene and Epidemiology, 4 Kozielska Str., 01-163 Warsaw, Poland

**Keywords:** camptothecin, biodegradable polyesters, bioresorbable polyesters, drug delivery systems, targeted therapy, biodegradable carriers, synthetic derivatives of camptothecin

## Abstract

Camptothecin (CPT) has demonstrated antitumor activity in lung, ovarian, breast, pancreas, and stomach cancers. However, this drug, like many other potent anticancer agents, is extremely water-insoluble. Furthermore, pharmacology studies have revealed that prolonged schedules must be administered continuously. For these reasons, several of its water-soluble analogues, prodrugs, and macromolecular conjugates have been synthesized, and various formulation approaches have been investigated. Biodegradable polyesters have gained popularity in cancer treatment in recent years. A number of biodegradable polymeric drug delivery systems (DDSs), designed for localized and systemic administration of therapeutic agents, as well as tumor-targeting macromolecules, have entered clinical trials, demonstrating the importance of biodegradable polyesters in cancer therapy. Biodegradable polyester-based DDSs have the potential to deliver the payload to the target while also increasing drug availability at intended site. The systemic toxicity and serious side-effects associated with conventional cancer therapies can be significantly reduced with targeted polymeric systems. This review elaborates on the use of biodegradable polyesters in the delivery of CPT and its analogues. The design of various DDSs based on biodegradable polyesters has been described, with the drug either adsorbed on the polymer’s surface or encapsulated within its macrostructure, as well as those in which a hydrolyzed chemical bond is formed between the active substance and the polymer chain. The data related to the type of DDSs, the kind of linkage, and the details of in vitro and in vivo studies are included.

## 1. Introduction

Despite significant advances in anticancer therapy, cancer remains the leading cause of death in developed countries. According to the American Cancer Society, there will be 1.9 million new cancer cases and over 609 thousand cancer deaths in 2022 [1]. The efficacy of therapy varies considerably depending on the type of cancer. Traditional chemotherapy is still a popular treatment option. Unfortunately, due to its nonspecific distribution, it does not produce optimal results. Chemotherapeutic agents inhibit rapidly growing malignant cells, but they also damage normal cells, particularly those with the highest rates of proliferation, such as hair follicles, bone marrow, and gastrointestinal tract cells. As a result, the most common side-effects associated with anticancer chemotherapeutic drugs are alopecia, anemia, weakened immunity, nausea, and vomiting [2,3,4,5,6,7]. Furthermore, the rapid clearance of anticancer drugs prevents the drug from reaching its therapeutic concentration, resulting in extremely high dose requirements. Furthermore, the vast majority of systemic chemotherapy drugs are lipophilic and easily absorbed by the liver. As a result, up to 85% of systemic chemotherapy patients develop hepatic steatosis [8]. Another disadvantage of traditional chemotherapy that should be emphasized is the development of multidrug resistance. This can be acquired by tumorous tissue in a variety of ways, including changes in the tumor microenvironment, the formation of alternative signaling pathways, or changes in target proteins [9].

Some of the abovementioned negative effects may be mitigated with modern targeted therapy. In order to improve the efficacy and safety of the treatment, the new approach assumes drug delivery to the target tissue and its controlled release.

New drug delivery systems (DDSs) include polymeric nanoparticles (NPs), liposomes, dendrimers, polymeric micelles, polymeric conjugates, or carbon nanotubes [2,10,11,12,13,14,15]. These DDSs enable the precise delivery of active substances to the site of tumor cells while avoiding healthy tissues. They also protect pharmaceuticals from rapid degradation, increase their half-life and payload, improve their solubility, and limit renal drug elimination [2].

As we mentioned above, conventional chemotherapy lacks target selectivity and frequently results in severe side-effects, limiting its effectiveness. As a result, innovative DDSs that ensure selective drug release and efficient intracellular uptake at the target sites are in high demand in order to improve the quality of life of patients while minimizing toxicity. Surface modifications (single or multiple) with various functional ligands such as transferrin, peptides, monoclonal antibodies, folic acid, hyaluronic acid, aptamers, anisamide, and biotin can be used to improve drug target selectivity. Furthermore, cell-penetrating peptides (CPPs), cholesterol, and ligands for tight junction opening in tumors are being actively pursued to improve the intracellular delivery of anticancer drugs [16]. These ligands can interact with specific receptors overexpressed in cancer cells, increasing drug uptake and ultimately making the therapy more effective and less toxic [16,17].

Camptothecin (CPT) is a cytotoxic alkaloid that inhibits cancer cell replication by interfering with the DNA topoisomerase 1 (Top1) [18]. Despite having significant antiproliferative efficacy, application of this drug is limited. The primary disadvantages of using CPT are its limited water solubility, lactone ring instability, poor biocompatibility, tumor cell resistance, and significant toxicity [19]. One of the possibilities for improving CPT characteristics is the development of novel its synthetic or semisynthetic analogues that improve its physicochemical properties, as well as its pharmacokinetic and pharmacodynamic profiles [20]. Furthermore, different physical and chemical methods for covalent or noncovalent association of CPT with various DDSs can be used as new opportunities for targeted therapy [20]. CPT encapsulated in polymeric NPs currently represents one of the most promising approaches for anticancer therapy due to the nanoscale size and high specific surface area [21]. The primary benefit of physical drug incorporation is increased permeability and retention impact, which promotes the transportation of NPs into the tumor environment. As a result, significant cellular absorption occurs, resulting in chemotherapeutic accumulation inside cancer cells [22]. Another alternative is to covalently link a drug to the macromolecular carrier to generate polymer–drug conjugates (PDCs). The drug’s pharmacological properties are then altered. Controlled delivery and drug release systems can be created using biodegradable polyesters; this enhances the precision of drug delivery to the tumor tissue and pharmacokinetics while reducing drug toxicity [23].

Multiple delivery systems have been used with different biodegradable and/or bioresorbable polyesters for anticancer therapy, taking into account the advantages of polymeric targeted DDSs over free drugs. This review aims to provide distinctive coverage of the field of biodegradable polyesters in the delivery of CPT and its analogues, as well as the use of these carriers in a variety of application strategies.

## 2. Camptothecin and Its Analogues

### 2.1. Camptothecin

*Camptotheca acuminata* (CPT, Chinese happy tree) is an indigenous tree found in China and Tibet, from which CPT was originally extracted. CPT ((*S*)-4-ethyl-4-hydroxy-1H-pyrano [3′,4′:6,7]indolizino [1,2-b]quinoline-3,14-(4H,12H)-dione) is an alkaloid found in all parts of the plant that contributes to the plant’s herbicidal defense mechanisms [24]. CPT (Figure 1 and Figure 2) is composed of three fused rings of pyrrolo-(3,4-b)-quinoline (rings A, B, and C), which are integrated with a pyridone (ring D) to form a planar pentacyclic ring structure. CPT in its active form contains a chiral center within the α-hydroxy lactone ring (ring E), which has an (S)-configuration and a delocalized aromatic moiety (rings A and B) [25].

CPT was isolated from the tree’s bark in 1966 and has since been used in Chinese traditional medicine to treat common colds, gastrointestinal disorders, and psoriasis [19]. This drug was discovered to be a Top1 inhibitor in 1985. Furthermore, it was revealed that CPT prefers inhibition of the Top1–DNA complex over inhibition of free Top1 enzyme [26]. Top1 enzyme expression in cancer cells is significantly higher than in healthy cells, allowing CPT targeted selectivity [25]. Despite CPT’s broad therapeutic potential, its effectiveness is limited by its low water solubility (2.5 × 10^−3^ mg/mL), rapid hydrolysis of the lactone ring in vivo, significant toxicity to mammalian cells, and acquired resistance [27,28]. At physiological pH, the lactone ring hydrolyzes, resulting in an equilibrium of the carboxylate, inactive pharmacologically form, and the active lactone form (Figure 1). The inactive carboxylate form has a high affinity for human serum albumins, which contributes to limited cellular uptake [29]. As a result, investigations into improving CPT water solubility have begun. Converting CPT to CPT sodium carboxylate salt was the first method. Unfortunately, the salt was ineffective and was excreted through the kidneys, causing myelosuppression, gastrointestinal toxicity, and hemorrhagic cystitis [27]. These findings emphasized the importance of developing novel CPT derivatives with enhanced pharmacokinetic and pharmacodynamic profiles [27]. The structure–activity relationship (SAR) (Figure 2) was crucial in the development of synthetic and semisynthetic CPT analogues, allowing the synthesis of molecules with improved properties when compared to natural drug [30]. The most important component of the CPT molecule is its planar pentacyclic ring structure, which is responsible for its anticancer activity [25]. However, anticancer activity has also been observed in derivatives with hexacyclic ring structures. This could imply that CPT needs at least five rings to continue acting as a Top1 inhibitor [31]. Furthermore, the A and B aromatic rings (Figure 2) are required because saturation of these rings results in poor activity even at high concentrations [25]; CPT loses all of its activity when the D ring is substituted with benzene [32], whereas ring-opening hydrolysis of the E-ring results in a significant reduction in anticancer activity. Lastly, the absolute configuration of C-20 atom is important for activity because the (S)-enantiomer is more active than the (R) [25].

### 2.2. Synthetic Analogues of Camptothecin

Topotecan (Figure 3 (1)), a CPT analogue, was the first Top1 inhibitor approved by the US Food and Drug Administration (FDA) [20]. It has been authorized for use as a second-line chemotherapeutic agent in the treatment of ovarian and small-cell lung carcinoma [24]. This CPT derivative was developed in order to improve the solubility and pharmacokinetics of the pristine compound. Topotecan has a basic amine side-chain at C-9, which makes it susceptible to the formation of ammonium salts and improves its water solubility at physiological pH [19]. Unfortunately, the modification of the quinoline structure resulted in a significant decrease in the compound’s cytotoxic activity, severely limiting its therapeutic use. The FDA also approved irinotecan (CPT-11) (Figure 3 (2)), which is used to treat large intestine cancer [20]. Its aqueous solubility was achieved by incorporating a basic side-chain at C-9 [19]. CPT-11 is converted in the body by the hepatic carboxylesterase enzyme into 7-ethyl-10-hydroxycamptothecin (SN-38). SN-38 is more cytotoxic than native CPT-11, but it is less water-soluble [20]. Another CPT derivative, rubitecan (9-CPT) (Figure 3 (3)), was created by adding a nitro group to the ninth position of the A ring of CPT. This analogue is also insoluble in water and has lactone ring instability [33]. Rubitecan is not only available orally, but also has the potential for transdermal or inhalation delivery. In equilibrium, it exists as 9-aminocamptothecin (9-AC) and 9-nitrocamptothecin (9-NC), both of which contain a lactone ring. The level of activity against human tumors observed in preclinical trials has not been replicated in clinical trials. Nonetheless, a promising activity against pancreatic cancer and possibly ovarian cancer has been described, with future clinical trials required [34].

Belotecan (CKD-602) (Figure 3 (4)), on the other hand, has recently been shown to have an antitumor effect in cervical cancer in both in vitro and in vivo models. CKD-602 increases the expression of the enzyme poly(adenosine diphosphate-ribose) polymerase (PARP), cleaved PARP, and Bcl-2-associated X protein (BAX), all of which are involved in cell apoptosis. Furthermore, the expression of phosphorylated p53 protein, which is involved in tumor suppression mechanisms, was increased. Following the treatment with belotecan, a significant reduction in cervical tumor volume was observed in this in vivo model [35].

Lurtotecan (GG-211) (Figure 3 (5)) is another water-soluble semisynthetic analogue of CPT that acts as a Top1 inhibitor. It has greater in vivo potency than topotecan. In preclinical studies, the antitumor activity was confirmed. Several phase I trials have demonstrated lurtotecan’s dose-limiting toxicity, with myelosuppression causing primarily neutropenia [36].

The encapsulation of GG-211 in low-clearance liposome preparations (OSI-211, NX-211) improved the drug’s preclinical pharmacokinetics, biodistribution, and therapeutic index. In preclinical studies, OSI-211 significantly increased plasma residence time, plasma area under the curve concentration (AUC) (1500-fold), drug accumulation in solid tumors (9–67-fold), and therapeutic index (3–14-fold) compared to GG-211. According to mouse models of human acute myeloid leukemia (AML) and acute lymphoblastic leukemia (ALL), OSI-211 exhibits significant antileukemic activity and is being investigated as a potentially new active agent for the treatment of leukemia [37].

Exatecan (DX-8951f) (Figure 3 (6)) was synthesized in 1995 as a novel CPT analogue. It is water-soluble, unlike CPT-11 and rubitecan. DX-8951f inhibits Top1 more effectively than natural CPT, topotecan, and SN-38 [24,38]. Furthermore, DX-8951f has been shown to have antineoplastic efficacy against a variety of cell lines, including lung, ovarian, cervical, colon, renal, and human breast [38,39]. Table 1 compares the selected properties and clinical trial status of CPT and its selected analogues.

## 3. Biodegradable Polyesters in Drug Delivery Systems

Biodegradable and/or bioresorbable polymers including polyesters should not cause systemic, immunologic, cytotoxic, mutagenic, cardiogenic, or teratogenic reactions, when applied in vivo. To the family of these materials belongs polylactide (PLA), which has emerged as a common biomaterial owing to its sustainable, biocompatible, and fully degradable properties [51]. To date, sustainable PLA has been increasingly used in biomedical materials and disposable commodities (e.g., food packaging). PLA comprising poly(D-lactide) (PDLA) and poly (L-lactide) (PLLA) enantiomeric polymers can create stereocomplex crystals through resume blending [52]. Multiple studies have recently identified stereocomplex-PLA (sc-PLA) as a promising carrier in various DDSs. Therapeutics can be encapsulated in this material using micelle formation, self-assembly, emulsion, or inkjet printing [53]. Poly(ε-caprolactone) (PCL) is another biodegradable polymer that can be used to encapsulate various types of drugs. PCL can be used to produce nanocapsules that can control drug release and improve a drug’s photochemical stability. Furthermore, because of their light-scattering capability, they can modulate cutaneous drug penetration and even act in protecting against physical UV radiation [54]. It is also worth noting that chemotherapy agents can be successfully delivered to the tumor site using PCL nanoplatforms, resulting in improved drug localization and antitumor efficacy while minimizing systemic side-effects [55]. Another polyester that is worth mentioning is polyglycolide or poly(glycolic acid) (PGA), which is a biodegradable, thermoplastic polymer and the simplest linear, aliphatic polyester [56]. PGA is degraded by enzymes and is highly hydrolytic in water with high pH ≥10. However, at near-neutral pH levels, PGA’s hydrolytic capacity is significantly reduced [57]. PGA and its copolymers are currently widely used as materials in the fabrication of absorbable sutures [58,59]. Among the polyester family, the copolymer of lactide (LA) and glycolide (GA) (PLGA) has been recognized as an important biocompatible and nontoxic polymer due to its biodegradability, biocompatibility, and sustained-release properties [60]. In the human body, PLGA is hydrolyzed nonenzymatically to produce mainly biodegradable monomers: lactic and glycolic acids. These compounds undergo biochemical reactions via the Krebs cycle and are then eliminated as carbon dioxide and water, resulting in minimal systemic toxicity [61]. Another biodegradable polymer worth mentioning is the copolymer of LA and ε-caprolactone (CL) (PLACL). Controlled drug-release properties, degradation rate, mechanical features, and shape-memory quality can all be regulated using PLACL, which are essential in the formulation of DDSs [62]. PLACL combines the mechanical hardness of PCL with the fast degradation rate of PLA, making it an excellent material for membrane and scaffold development [63]. PLACL could also be used in the preparation of nerve guides due to its exceptional properties for providing guidance and a protective barrier for nerve fiber regeneration [64]. Last but not least, the copolymer of GA and CL (PGACL) is a widely used biodegradable and bioresorbable copolymer. PGACL is broadly used in the development of shape-memory materials, implants, and scaffolds. Terpolymers of GA, LA, CL, and trimethylene carbonate (TMC) are also known as biomaterials [65,66,67]. PGACL is a suitable material for the design of new DDSs and medical devices because it is stable in mechanically dynamic environments and promotes appropriate cellular interactivities [65]. Poly(3-hydroxybutyrate) (PHB), on the other hand, was initially investigated as a biodegradable packaging material [68]. PHB has a high crystallinity, a relatively high melting point, and adequate hydrolytic stability. These characteristics allow it to be used for biomedical purposes. Its copolymers can be used in bone implants; however, in the last decade, PHB-based nanoparticles have also been considered as promising DDSs [69]. This polymer is not only biodegradable, but also bioresorbable, which means that it can be eliminated naturally through filtration or metabolism [70]. Currently, biodegradable DDSs, including drug-loaded nanoparticles, are generally prepared using the polymeric materials mentioned above, particularly with poly(ethylene glycol) (PEG). PEGylation of particles is a promising method for extending their lifetime in the bloodstream [71,72].

## 4. Polyester Carriers of Camptothecin and Its Analogues in Cancer Therapy

As previously noted, CPT was found to be ineffective when administered to patients, owing primarily to the hydrolysis of its lactone ring at physiological pH. Furthermore, CPT is characterized by low solubility, high toxicity, and rapid inactivation in the bloodstream [23]. Nanomedicine offers a solution to these disadvantages while also delivering CPT and its analogues to target cells in a safe and effective manner. As an example, consider the physical incorporation of CPT or CPT derivative into the structure of a polymeric carrier, or the chemical conjugation of the drug to the polymer chain via a chemical bond [20]. Paying attention to the first, we can mention encapsulation or absorption of the active substance onto polymer surface. Although various polymers have been used for drug encapsulation or adsorption, only biodegradable and biocompatible polymers are suitable for biomedical applications. Correspondingly, a covalent bond may be formed between CPT or CPT derivative and polymer macromolecule, thus creating polymer–CPT conjugates [73]. Helmut Ringsdorf was the first to propose a drug–polymeric carrier conjugate model in 1975, which included a biocompatible polymer backbone to which three components were connected: (1) a solubilizer or modifier that provides hydrophilicity and water solubility, (2) a drug that is covalently linked to the polymeric backbone, and (3) a targeting moiety that attracts and transports the carrier–drug conjugate to a specific physiological destination or biological target [15]. There are numerous forms of CPT delivery available, which can be classified into specific groups on the basis of their design and chemical nature [20].

### 4.1. Encapsulation and/or Adsorption of Camptothecin and Its Analogues onto a Polyester Carrier

DDSs can be created using a variety of polymers. Collagen, albumin, gelatin, alginate, cyclodextrin, chitosan, starch, and cellulose are examples of natural macromolecules. On the other hand, various biodegradable polyesters, such as PLA, PCL, or copolymers of LA, CL, or GA (e.g., PLGA), might be synthesized. The advantage of synthetic polymers over natural ones is the ability to obtain carriers with a specific microstructure, as well as physicochemical, mechanical, and thermal properties for a variety of applications. The disadvantages of natural polymers include the possibility of microbial contamination, a high level of variability, and expensive extraction processes, as well as a lack of control over hydration [74]. Different types of drug-delivery structures can be created using biodegradable polyesters. Polymeric micelles (20–80 nm) comprise amphiphilic block copolymers that combine to form a spheroidal structure which can hold hydrophobic anticancer drugs. The hydrophilic shell, on the other hand, ensures the structure’s stability. Polyester dendrimers (10–100 nm) are highly branched three-dimensional macromolecules that can cross cancer cell membranes and reduce the clearance by macrophages. The anticancer drug can be physically encapsulated in the dendrimer’s core or covalently conjugated to its surface. Polymeric NPs are colloidal systems containing nanosized particles, in which drugs can be encapsulated into a space restricted by a polymeric carrier (nanospheres) or entrapped into a cavity surrounded by a polymeric membrane (nanocapsules) [2]. Polymeric NPs have several benefits as drug carriers, including the possibility for controlled release and precise transport, the capacity to protect drugs from the environment and other molecules with biological activity, and the improvement of their bioavailability and therapeutic index. Liposomal DDSs provide stable formulation and improved drug pharmacokinetics. Liposomal anthracyclines, for example, contain drugs that are encapsulated with high efficiency, resulting in a significant reduction in cardiotoxicity and prolonged circulation. After reaching the tumor tissue, the drug-loaded liposomes remain in the tumor stroma until enzymatic degradation occurs, at which point the anticancer drug is released and acts in the desired area [75].

At this stage, it is worthwhile to start discussing drug loading techniques. All drug loading strategies are focused on either covalent or noncovalent drug systems. Covalent approaches involve chemically bonded tethers, whereas noncovalent drug delivery includes all other drug delivery methods, such as hydrophobic, electrostatic, hydrogen bonding, and steric immobilization [76]. Additionally, these widespread strategies can be further classified into surface-mediated or encapsulation-based procedures [77]. Notably, taking into account these variations, both covalent and noncovalent drug delivery strategies demand the optimum loading sites and the minimal hindrance to diffusion from the NP vector after it has reached the target site. Both of these similarities depend on the particle, and a favorable environment near to the NPs is necessary to facilitate drug loading. Physical forces in the local environment may have a far higher effect on noncovalent drug distribution. Inorganic NP-mediated encapsulation and surface-mediated noncovalent drug loading are frequently investigated in the literature. Understanding the dominant physical forces in NP–drug interactions is essential for optimizing drug delivery. The loading of therapeutics noncovalently is always influenced by a number of physical factors, which together make up the system’s total cooperative force. Both noncovalent and covalent drug loading strategies require effective drug attachment to the NP carrier in order to attain therapeutic efficacy. For in vivo applications, a higher drug loading efficiency is often preferable; however, with improved drug development, these requirements may be relaxed for the treatment of a specific illness [77].

Since the 19th century, hydrogels have been described in the literature. Their biomedical applications have advanced rapidly over the time. These materials are currently defined as crosslinked polymer systems that form a three-dimensional structure capable of absorbing water, body fluids, nutrients, metabolites, or small hydrophilic molecules while remaining integral. Because of their ability to ionize in an aqueous environment, hydrophilic functional groups play an important role in the hydrogel structure [78]. Nanogels, which are hydrogel particles, can also be used for drug encapsulation. These carriers are typically composed of hydrophobic polysaccharides with high water absorption, versatility, and biocompatibility. As a result, nanogels are used for drugs that are poorly soluble in water. The hydrogel release system is either time-controlled or stimulus-responsive. The stimuli such as temperature, electricity, pressure, sound, or light can trigger and activate drug release [79].

The primary benefits of polyester DDSs produced through drug encapsulation or adsorption are usually increased water solubility, biocompatibility, extended drug tolerance time, precise delivery and accumulation at the site of action, and reduced toxic side-effects on unimpaired cells [80]. Unfortunately, one of the major issues with polymeric DDSs characterized by controlled drug release, particularly micro and NPs, is the initial burst [81]. The drug’s initial burst release is a phenomenon that occurs when a large amount of the drug is liberated before it can reach a stable release rate [82]. This is especially dangerous when toxic drugs, such as CPT, are administered. The difficulty of burst release can be solved by adjusting the drug distribution within the polyester carrier or by establishing more refined DDSs [81].

Importantly, NP surfaces can be modified to deliver drugs to precisely targeted cancer tissues. This surface modification also prevents adverse interactions between NPs and blood morphological elements [83]. Recently, researchers have focused on increasing the site-specific activity of polymeric DDSs by coupling their structure with ligands that target specific antigens or receptors on the cell surface [84]. As an example related to the topic of this review, monoclonal antibodies (mAbs) have been linked to the surface of the particle in order to precisely direct DDSs to the tumor environment. Because the combination of antibodies and nanodrugs can improve treatment efficacy, antibody-conjugated polymeric prodrug NPs for targeted CPT were developed. After attaching CD147 monoclonal antibody (CD147 mAb) to the NPs, it was discovered that the antibody specificity allows it to bind to the CD147 protein (transmembrane glycoproteins that are highly expressed on the surface of epithelial tumor cells, such as lung cancer, breast cancer, and liver cancer), which is overexpressed in human hepatocellular carcinoma cells (HepG2). Endocytosis tests revealed that CD147–CPT NPs had higher uptake rate and accumulation in HepG2 cells than CPT-loaded NPs without antibodies, owing to the fact that CD147 mAb can specifically bind to the CD147 protein, which is overexpressed in HepG2 cells. The authors developed a method for attaching monoclonal antibodies to anticancer polymeric prodrugs and endowed biodegradable polymeric prodrugs with precise targeting functions to liver cancer cells [17].

In Table 2, we demonstrate polyester DDSs of CPT and its analogues achieved through drug encapsulation or adsorption. This table includes both in vitro and in vivo detailed information. However, we mention and explore some of the most inspiring examples, focusing on the findings of in vitro and in vivo studies.

Padhi and coworkers concentrated on the entrapping of topotecan hydrochloride into a biodegradable PLGA matrix in order to obtain topotecan NPs. Topotecan-loaded PLGA NPs were generated using a double-emulsion solvent evaporation technique. The statistical optimization of the process using the Box–Behnken design yielded NPs with a size of 243.2 ± 4 nm, zeta potential (ζ) of −2.36 ± 0.6 mV, and entrapment efficiency (*EE*) of 60.9% ± 2.2%. A sustained drug release from the formulated PLGA NPs was observed for over 1 week in phosphate-buffered saline (PBS) at both physiological (pH 7.4) and tumor microenvironmental conditions (pH 6.5). Furthermore, the biological study’s findings revealed enhanced cellular uptake by human ovarian cancer cells (SKOV-3) over time and 13-fold higher bioavailability compared to the native drug. The supremacy of passively targeted topotecan NPs was further demonstrated by the pharmacokinetic results, which revealed a 13.05-fold increase in bioavailability in the biological system [85].

An interesting study was described by Yang et al. which investigated the potential of CPT-11-loaded PLGA NPs produced by an emulsion–solvent evaporation method. The obtained NPs were spherical, with an average size of 169.97 ± 6.29 nm, *EE* of 52.22 ± 2.41%, and drug loading (*DL*) of 4.75 ± 0.22%. The in vitro release characteristics were then investigated. The results showed that the irinotecan-loaded PLGA NPs could continuously release the drug for 14 days. For the pharmacokinetic and pharmacodynamic studies, Kunming mice were used as an animal model. In pharmacokinetics investigations, for the studied drug, the half-life (t_1/2_β) of CPT-11 loaded PLGA NPs was extended from 0.483 to 3.327 h compared with CPT-11 solution; for its active metabolite SN-38, the t_1/2_β was extended from 1.889 to 4.811 h, indicating that CPT-11-loaded PLGA NPs could prolong the retention times of both irinotecan and SN-38. The pharmacodynamics results revealed also that the tumor doubling time, growth inhibition rate, and specific growth rate of CPT-11-loaded PLGA NPs were 2.13-, 1.30-, and 0.47-fold those of the CPT-11 solution, respectively, demonstrating that CPT-11 loaded PLGA-NPs could significantly inhibit growth of the tumor [86].

In continuity, Tseng and coauthors established SN-38-loaded PLGA microparticles (SMPs) using the electrospraying technique for applications in the treatment of central nervous system pathologies (malignant glioma; MG). The measured particle size and ζ value were 1.58 ± 0.54 µm and −0.86 ± 0.10 mV, respectively. The levels of SN-38 eluted from the microparticles were determined using an in vitro elution method. The biodegradable SMPs delivered high SN-38 concentrations for more than 8 weeks while causing temporary inflammation in brain tissue. The F98 MG-bearing rats treated with SMPs benefited from restricted and retarded tumor growth, extended survival, and reduced malignancy. When used at concentrations lower than those causing dose-limiting systemic or neurological toxicity to the normal brain, the produced material demonstrated favorable activity against MG in F98 MG-bearing rats. The findings suggested that SMPs have the potential to be effective for interstitial chemotherapy and are a viable alternative to the current therapeutic options for MG [87].

Last but not least, Ci et al. in their work reported a formulation containing CPT-11 and a PLGA–PEG–PLGA copolymer. The PLGA–PEG–PLGA aqueous solution was a sol at room temperature and physical gel at body temperature, forming a thermogel. CPT-11 was successfully incorporated into the amphiphilic copolymer aqueous solution. In vitro studies revealed that CPT-11 was released from the thermogel over a 2 week period. The mixture was injected subcutaneously into nude mice with xenografted SW620 human colon tumors. The animal group that received the CPT-11-loaded thermogel demonstrated excellent in vivo antitumor efficacy. The tumor regressed significantly after being treated with CPT-11/thermogel, and the side-effects (blood toxicity and weight loss) were minimal. These findings could be attributed to the thermogel’s ideal sustained-release profile and period of release of the drug, as well as the thermogel’s significant enhancement of the fraction of the active form of the drug [88].

**Table 2 ijms-24-01053-t002:** Encapsulation and/or adsorption of CPT and its analogues to form different types of biodegradable polyester DDSs.

Type of DDSs	Drug	Polyester Carrier Used for the Formulation	Size, Zeta Potential (ζ) and Entrapment Efficiency (EE)	In Vitro Studies				In Vivo Studies				Additional Studies	Reference
				Release Date	Cytotoxicity	Cellular Uptake	Cell Line	Pharmacokinetics Study	Animal Model	Pharmacodynamics Study	Cell Line		
Nanoparticles	Topotecan	PLGA	243.0 ± 4.0 nm ζ = −2.36 ± 0.6 mV EE = 60.9% ± 2.2%	Release rate after 1 day: was 25.06% ± 2.4% at pH 6.5 and 22.45% ± 2.4% at pH 7.4.	${\mathrm{IC}}_{50}=1.8\;\pm \;$0.98 (after 48 h) ${\mathrm{IC}}_{50}=1.2\;\pm$ 0.43 (after 72 h)	87.4% ± 2.6% within 1 h	SKOV3	Noncompartmental method: Cmax=1326 ± 17.89 ng/mL AUC_0–∞_ = 98,978.15 ± 362 ng h/mL Tmax=3 ± 0.3 h	Swiss albino mice	-	SKOV3	-	[85]
	9-CPT	PLGA	207 ± 2.6 nm	At pH 7.4, 20% of the drug was released in 20 h	At concentration of 5 µg/mL, after 24 h, percentage of cell viability was equal to approximately 10%. After 24 h, 91% of the cells were killed	Time-dependent cellular uptake, increased with time: about 40 μg/mL after 3 h	A2780sn—cytotoxicity study Caco-2—cellular uptake study	Noncompartmental method (total 9-CPT): ${\mathrm{AUC}}_{0–\infty }=3692\;\pm \;$868 ng h/mL T1/2 = 2.45 ± 0.27 h MRT = 1.56± 0.36 h Vss=195 ± 59 mL	Male Wistar rats	-	-	Empty NPs were nontoxic	[89,90,91]
	CPT-11	PLGA	124 ± 12 nm ζ = −20.3 mV EE = 55% ± 2.7%	At pH 7.4, about 50% of the drug was released within 24 h	IC_50 = 36.2_ ± 1.2 (after 48 h)	-	HT-29	-	-	-	-	-	[92]
	CPT-11	PLGA	169.97 ± 6.29 nm ζ = −0.94 ± 0.6 mV EE=52.22% ± 2.41 %	At pH 7.4, 21.43% ± 1.3% of the drug was released within 2 h	-	-	-	Compartmental method: AUC_0–∞_ = 16.8 mg/L·h t1/2β = 3.327 h	Kunming mice	Tumor growth inhibition (TGI) = 86.63%	H22	≤5% hemolysis of CPT-11 PLGA NPs at CPT-11 concentration 20–100 µg/mL	[86]
	CPT-11	PCL	202.1 ± 2.1 nm ζ = ca. − 8.0 mV EE = 65%	40.4% ± 1.5% of the drug was released within 16 days	CPT-11-NPs were significantly more cytotoxic by day 11 compared to day 1 due to the slow release of IRH over the 11 days	-	Primary HGG	-	-	-	-	-	[93]
	SN-38	PLGA–PEG–FOL	221 ± 15 nm ζ=−6.5 ± 0.3 mV EE=89.1% ± 9.2%	At pH 7.4, about 23% of the drug was released within 240 h	$\mathrm{Calculated}\;{\mathrm{IC}}_{50}\;\mathrm{value}\;\mathrm{was}\;30\%\;\pm \;$1.1% lower than that of nontargeted NPs (after 48 h)	Higher cellular uptake than PLGA-NPs	HT-29	-	-	-	-	-	[94]
	SN-38	PLGA	173 ± 13 nm ζ=−10.8 ±0.2 mV EE = 77.1% ± 6.5%	At pH 7.4, about 30% of the drug was released within 240 h	${\mathrm{IC}}_{50}\;\mathrm{was}\;\mathrm{calculated}\;\mathrm{as}\;51.5\%\;\pm \;2.3$%	-	HT-29	-	-	-	-	-	[94]
	SN-38	PLGA	282.9 ± 24.5 nm ζ=−11.3 ± 4.1 mV EE=70.5 ± 14.9	At pH 5.0, about 55% of the drug was released within 15 days; at pH 7.4, about 26% of the drug was released within 15 days	IC _50_ = 0.874 μM (after 48 h)	Active targeting and prolonged circulation properties	CD44 Her2 HGC27	-	Balb/c nude mice	-	-	In vitro anti-proliferation mechanism revealed downregulated expression of CD44 and Her2 and better inhibition of HGC27 cell growth and invasive activity	[95]
	SN-38	PLGA–PEG–PLGA (70,000:8000:70,000) (HMw) PLGA–PEG–PLGA (6000:10,000:6000) (LMw)	HMw in PBS pH 7,4: 73.43 ± 0.25 nm ζ=−15.73 ± 2.20 mV EE = 6% LMw in PBS pH 7.4: 56.06 ± 0.40 nm ζ=−8.53 ± 1.39 mV EE = 3%	-	Negligible growth inhibition effect. Cell viability approximately 90%	Increasing cellular uptake over time, during 24 h internalization study of fluorescently labeled NPs	SW-480	-	Wistar rats	-	-	Increased expression of UBD and RGCC genes. Decreased expression of FGF3 and HIST genes. HMw-NPs accumulated rapidly in the liver. LMw-NPs were detected 1 h post injection. After 24 h LMw-NPs were mostly distributed in the liver	[96]
	CPT	PLGA	128.4 ± 8.6 nm ζ=−11.7 ± 0.4 mV EE=64.8% ± 5.5%	54.7% ± 4.2 % of the drug was released within 12 h. 87.3 ± 7.4% of the drug was released within 48 h	At concentration 150 µg/mL, after 48 h cell viability was suppressed by 46.5% ± 4.8% compared with CPT	Concentration-dependent endocytic process: higher cellular uptake after 4 h incubation than free CPT at 150 µg/mL.At 50 and 100 µg/mL, there were no significant differences	HepG2	-	-	-	-	Functional study of CYP3A4 activity revealed that the activity of the cytochrome P450 may be inhibited by CPT-PLGA NPs	[97]
Micelles	SN-38-BOC	MPEG-P(CL-ran-TMC)	33 ± 1 nm ζ = −0.6 mV EE=98.6% ± 1.1%	28.3% ± 1.3% of the drug was released within 24 h	IC_50_ = 2.1 µg/mL (after 48 h)	-	HCT116 CT26	-	Female BALB/c mice Female BALB/c nude mice	Significant difference in tumor volume and weight. TGI for SN-38-BOC micelles=80.5% ± 2.4%	HCT116 CT26	SN-38-BOC micelles and free SN-38-BOC inhibited embryonic angiogenesis in transgenic zebrafish embryos. Mice administered with SN-38-BOC micelles maintained body weight	[98]
Microspheres	CPT-11	PLA	37.2 μm EE = 93.4%	About 70% of the drug was released within 1 day	-	-	-	-	-	-	-	-	[99]
Nanocapsules	CPT-11	PLGA	103.4 nm EE ≈ 65%	At pH 7.4, about 20 % of the drug was released within 5 days. At pH 5.5, about 30% of the drug was released within 5 days	Material inhibited cell survival for both lines.	-	SW1990 Panc-1	-	BALB/c (nu/nu) nude mice	No significant difference in tumor volume within 5 days after administration. By day 15, the tumor volume was smaller than that of the free drug (*p* ˂ 0.001)	-	-	[100]
Hydrogels	MPEG–CPT	PLGA–PEG–PLGA	-	In PBS about 70% of the drug was released within 35 days.	-	-		-	Mice	Tumor inhibition ratio = 73.1% after 3% (*w*/*w*) MPEG-CPT injection (c.a. 56 mg/kg)	Murine S180 sarcoma	-	[101]
	CPT-11	PLGA–PEG–PLGA	-	Sustained release throughout 2 weeks. About 50% of the drug was released within 4 days	In vitro cytotoxicity of PLGA–PEG–PLGA: more than 80% cell viability (10 mg/mL polymer concentration). In vitro hematoxicity of PLGA–PEG–PLGA: (2.8% at 2 mg/mL copolymer concentration). Weak inflammatory response after injection	-	MC3T3	-	Mice	Tumor inhibition ratio from 86.2% (CPT-11 in thermogel: 1 mg/mL) to 98.2% (CPT-11 in thermogel: 4 mg/mL)	Mice xenografted SW620 human colon tumors	Reduced side-effects in Kunming mice bearing murine solid tumor S180 (drug dose: 45 mg/kg)—lower WBC decrease and rapid WBC recovery in CPT-11 thermogel group than in the CPT-11 group. The body weight of nude mice—no marked difference between the experimental and tumor-free group was observed	[88]
Nanogels	CPT	PLA–PEG–PLA diacrylate	EGDMA 3% 237.56 ± 2.49 nm EE=85.4% ± 2.5% EGDMA 6% 178.29 ± 8.81 nm EE=84.1% ± 6.9% EGDMA 12% 166.31 ± 5.57 nm EE=83.1% ± 4.1% EGDMA 24% 157.20 ± 9.68 nm EE=82.1% ± 5.7% EGDMA 50% 157.41 ± 8.00 nm EE=81.7% ± 6.1%	In PBS, within 20 days: EGDMA 3%, about 7% of the drug was released. EGDMA 6%, about 4% of the drug was released. EGDMA 12%, about 3% of the drug was released. EGDMA 24%, about 2% of the drug was released. EGDMA 50%, about 1% of the drug was released	-	-	-	-	-	-	-	The size of nanogel did not change within 2 months of storage at 4 °C	[102]

t_1/2_—half-life; TGI—tumor growth inhibition; MRT—mean residence time; V_ss_—volume of distribution at steady stage.

### 4.2. Polyester Conjugates of Camptothecin and Its Analogues

Variable polymeric DDSs, including PDSs, have recently captured the attention of researchers. This strategy has been found to be capable of improving drug pharmacokinetic parameters, increasing drug stability against degradation, providing high loading capacity and sustained release patterns, and avoiding premature drug release. It is possible to deliver both hydrophobic and hydrophilic drugs using the polymer conjugate approach, which is typically difficult with drug-loaded NPs prepared via physical encapsulation with hydrophobic interaction as the main mechanism [20]. PDCs can be formed in three ways: (1) by incorporating a drug to a polymer carrier, (2) by incorporating a drug into a monomer prior to polymerization, and (3) by incorporating a drug as monomers or initiators during the polymerization reaction. The second method achieved controlled and high drug loading while not interfering with polymerization or hindering conjugation [103]. Several polymerization reactions were used to develop PDCs using the second method, and triggered drug release from conjugates loaded with multiple drugs has been obtained. PDCs with biodegradable backbones were invented using ring-opening polymerization (ROP). However, some PDCs with nonbiodegradable backbones have been produced using ring-opening metathesis polymerization (ROMP) and reversible addition/fragmentation transfer polymerization (RAFT) [104]. Furthermore, the drug can be linked to the functional groups of the polymer carrier directly or via a “bioresponsive” spacer. The spacer can respond to biological variables such as pH changes or the presence of specific enzymes. As a result, PDCs can be designed to support drug release at specific sites of action [105]. Many different types of bonds can link the active substance to the polymeric chain (Figure 4). Their hydrolysis or biodegradability is crucial for drug release [106].

As outlined, the rate of drug release is strongly influenced by the type of bond that links the drug to the polymeric backbone. The different types of bonds imply a conjugate’s susceptibility to enzymatic and/or hydrolytic degradation (for instance, ester bonds are more labile than amide bonds; thus, conjugates with ester bonds can degrade more quickly). This is why the drug can be released from the conjugate in the desired location when the appropriate conditions are present (e.g., acidic tumor microenvironment, tissue enzymes, and specific antibodies). Furthermore, the drug release may be controlled by the composition and topology of the polymeric carrier (e.g., PDLLA indicates a faster rate of degradation compared to PCL) and by its microstructural, physicochemical, and mechanical characteristics. Appropriate selection of these features results in the desired type of drug release systems [107].

To date, some efforts have been undertaken to obtain and characterize polyester conjugates of CPT and its analogues. Naturally, the materials synthesized may vary in the types of linkage between the pharmacologically active substance and the polymeric chain. Table 3 summarizes the biodegradable polyester carriers developed thus far and utilized to obtain conjugates with CPT or its analogues.

In a broader context, we would like to expand on the efforts of Sobczak and coworkers, where the authors prepared for the first time various macromolecular conjugates of CPT. The ROP of CL, GA, *rac*-lactide (*rac*-LA), or trimethylene carbonate (TMC) allowed obtaining different polymeric carriers, which were then conjugated with CPT via the urethane bond. The preliminary cytotoxicity study of the obtained copolymeric carriers was tested on *Vibrio fischeri* bacteria and two protozoans: *Spirostomum ambiguum* and *Tetrahymena thermophila.* The performed tests revealed that the obtained polyesters or polyester-carbonates were nontoxic to all test bionts. Furthermore, CPT release rates were investigated in vitro and found to be dependent on the nature of the obtained copolymers. In some cases, CPT was released from the obtained DDSs with a high controlled rate [23].

Another study conducted by Oledzka et al. focused on the synthesis of biodegradable (atactic PLA)_100_ and (atactic PLA)_50_-*b*-(isotactic PLA-*P*_m_ = 0.79)_50_ conjugates of CPT (Figure 5). The developed matrices were tested for cytotoxicity using the *Microtox^®^*, *Protoxkit F^TM^*, and *Spirotox* tests. The results showed that the synthesized polyester carriers were nontoxic to all test bions. CPT was conjugated to the synthesized matrices via a 1,6-diisocyanatohexane (HMDI) linker, resulting also in a urethane bond. The findings show that CPT release was strongly influenced by the polymer microstructure and pH environment. Furthermore, by modifying the microstructure of polymers this method may be a promising for developing polyester DDSs for anticancer drugs with prolonged and controlled release [108].

In the next example, a facile method based on the combination of Michael addition polymerization and CuAAC “click” chemistry was utilized to develop a new reduction-responsive polyphosphoester-based CPT prodrug P(EAEP–PPA)–*gss*–CPT. Drug derivatives with disulfide bonds and azido groups were linked to the carrier’s side-chain (Figure 6). This amphiphilic prodrug could self-assemble into micelles in aqueous solution, with the hydrophobic CPT segment as the core and the hydrophilic P(EAEP–PPA) segment as the corona; the micelles were 141 nm in size. The drug’s release was attributed to reduction-responsive behavior in the presence of glutathione (GSH); the rate of release was directly proportional to the GSH concentration used in the media. The cytotoxicity of the prodrug micelles was investigated using mouse breast cancer cells (4T1 cells) and human hepatocellular carcinoma cells (HepG2 cells). The results showed that the synthesized material was biocompatible and effectively inhibited the cell proliferation of 4T1 and HepG2 cell lines. Furthermore, the CPT prodrug micelles could be internalized into HepG2 cells to deliver active CPT via endocytosis [109].

On the other hand, Zhao and coauthors, developed multi-arm PEG conjugates of SN-38 (Figure 7). These materials maintained the active form of SN-38, preventing it from being converted to an inactive, carboxylate form. Amino-acid spacers (alanine, methionine, sarcosine, and glycine) were used to link SN-38 to PEG carrier, resulting in a highly water-soluble formulation. The synthesized conjugates were stable in neutral conditions and had a high release rate at pH levels above 6.0. The cytotoxicity assay, which used human colon cancer cell lines (COLO-205 and HT-29), human ovarian carcinoma cell line (OVCAR-3), and human lung adenocarcinoma cell line (A459), revealed high potency against these cell lines. Furthermore, in vivo studies on female nude mice with MX-1 tumors were carried out. Mice were injected intravenously with a single dose of 20 mg/kg or multiple doses of 5 mg/kg (every two days) of the synthesized conjugates. The treatment resulted in significant tumor growth inhibition (>99%) and no loss of body weight, implying that the obtained formulation is a promising anticancer agent [110].

The clinical application of SN-38 is limited due to its poor aqueous solubility and lactone ring instability at physiological pH. As a result, Lu and coworkers reported the synthesis of novel amphiphilic SN-38 conjugates capable of forming micelles through a self-assembly mechanism using methoxypoly(ethylene glycol)-*b*-poly(lactide) (mPEG–PLA), a typical amphiphilic block copolymer that acts as a surfactant. The polymeric carrier was reacted with SN-38 via an esterification reaction between the carboxyl group of the polymer and the phenolic hydroxyl group in SN-38. The resulting conjugate micelles (mPEG–PLA–SN-38) had a mean diameter of 13.4–1.2 nm and a ζ value ranging from 5.64 to 9.99 mV. The effects of mPEG–PLA composition were studied in vitro and in vivo, and it was discovered that mPEG2K–PLA–SN-38 conjugates were more effective against tumors than mPEG4K–PLA–SN-38. The authors concluded that the lengths of the mPEG and PLA chains had a significant impact on the physicochemical properties and antitumor activity of SN-38 conjugate micelles [111].

**Table 3 ijms-24-01053-t003:** Biodegradable polyester DDSs of CPT and its analogues obtained by covalent drug conjugation.

Polymer	Linkage	Drug	In Vitro Studies			In Vivo Studies	Cell Line Tested	Additional Studies	Benefits, Conclusions	References
			Release Kinetics	Cytotoxicity	T_1/2_					
(attacic-PLA)_100_ (PLA-1) and (attacic-PLA)_50_-*b*-(isotactic PLA-P_m_ = 0.79)_50_ (PLA-2)	Urethane bond	CPT	In buffer solution at pH 1.0, 20% of the drug was released after 35 days from the CPT-PLA-1 and 6% from the CPT-PLA-2 conjugate. In buffer solution at pH 7.4: CPT-PLA-1 led to 7% release, and CPT-PLA-2 led to 4% release after 35 days of incubation	-	-	-	-	PLA matrices were nontoxic according to Spirotox, Protoxkit, and Microtox tests	Drug-release characteristics were strongly influenced by the PLA microstructure chain	[108]
PGACL (initiator: PEG-200) PLACL (initiator: PEG-200) poly(CL-*co*-TMC) (initiator: PEG-200) PLGA (initiator: PEG-200) poly(GL-*co*-TMC) (initiator: PEG-200) poly(LA-*co*-TMC) (initiator: PEG-200) PLACL (initiator: PEG-400) PLACL (initiator: PEG-600)	Urethane bond	CPT	In buffer solution, at pH 7.4, after 12 weeks of incubation, 81%, 73%, 32%, 89%, 39%, 35%, 82%, and 89% of CPT were released, respectively	-	-	-	-	Synthesized matrices were nontoxic according to Spirotox, Protoxkit^F^, and Microtox tests	The rates of CPT release were shown to be directly dependent on the nature of the synthesized carriers. The kinetic rates of drug release were found to be faster in polymeric conjugates containing CL, *rac*-LA, or GL units compared to those containing TMC units. The rate of in vitro CPT release from the macromolecular conjugates increased as the *M*_n_ of PEG increased	[23]
P(EAEP–PPA)	Disulfide bond	CPT	In buffer solution at pH 7.4, about 20% of the drug was released within 1 day. In buffer solution at pH 7.4 + 2 μM GSH, about 20% of CPT was released within 1 day. In buffer solution at pH 7.4 + 5 mM GSH, about 35% of drug was released within 1 day. In buffer solution at pH 7.4 + 10 mM GSH, about 50% of the drug was released within 1 day	Synthesized polymers were nontoxic	-	-	4T1, L929, and HepG2	Efficient cellular uptake of CPT endocytosis	Efficient inhibition of 4T1 and HepG2 cell proliferation	[109]
PEG-3400	Ester bond	CPT	In PBS, about 90% of CPT was released within 10 days	-	-	-	-	Higher stability of the conjugate at pH 6.0 and 5.5 than at pH 7.4.	-	[112]
PEG-40000	Ester bond, amino acid spacers (alanine, methionine, sarcosine, and glycine)	SN-38	-	IC_50_, µM: COLO 205 (colorectal): 9, 0.13 ± 0.022; 13, 0.10 ± 0.024; 18, 0.18 ± 0.013; 23, 0.14 ± 0.032. HT29: 9, 0.21 ± 0.068; 13, 0.21 ± 0.049; 18, 0.34 ± 0.081; 23, 0.52 ± 0.066. OVCAR-3: 9, 0.22 ± 0.01; 13, 0.2 ± 0.047; 18, 0.27 ± 0.11; 23, 0.1 ± 0.032. A549: 9, 3.9 ± 0.97; 13, 2.1 ± 0.28; 18, 5.1 ± 0.95; 23, 3.1 ± 043	In human plasma: 9–12.5 min, 13–26.8 min, 18–19 min, and 23–12.3 minIn rat plasma: 9–6.3 min, 13–12.4 min, 18–10.5 min, and 23–3.5 min	Female nude mice–human mammary carcinoma (MX1) breast tumor	COLO 205, HT 29, A549, and OVCAR 3	Much enhanced anticancer activity of SN38 in the MX-1 xenograft mice model compared with CPT-11	-	[110]
CD-123	Disulfide linker	CPT	Release achieved in the presence of GSH	-	-	-	THP-1 and Hep3B	-	-	[113]
PEG_2K_–PLA_4-2K_	Ester bond	SN-38	In PBS at pH 7.4 with addition of 0.2% Tween-80, about 25% of the drug was released within 1 day. In PBS at pH 7.4 with addition of 20% (dialysis medium), about 45% of the drug was released within 1 day	Against BEL-7402: IC_50_ = 5.33 ± 0.84 μg/mL. Against HCT116: IC_50_ = 0.93 ± 0.18 μg/mL	-	-	BEL-7402 and HCT116	-	-	[111]
PEG_2K_–PLA_8-9K_	Ester bond	SN-38	In PBS at pH 7.4 with addition of 0.2% Tween-80, about 25% of the drug was released within 1 day. In PBS at pH 7.4 with addition of 20% dialysis medium, about 45% of the drug was released within 1 day	Against BEL-7402: IC_50_ = 2.35 ± 0.11 μg/mL. Against HCT116: ${\mathrm{IC}}_{50}=0.52\;\pm \;0.05\;$μg/mL	-	-	BEL-7402 and HCT116	-	-	[111]
PEG_4K_–PLA_1K_	Ester bond	SN-38	In PBS at pH 7.4 with addition of 0.2% Tween-80, about 25% of the drug was released within 1 day. In PBS at pH 7.4 with addition of 20% dialysis medium, about 50% of the drug was released within 1 day	Against BEL-7402: IC_50_ = 21.38 ± 2.44 μg/mL. Against HCT116: IC_50_ = 21.06 ± 1.19 μg/mL	-	-	BEL-7402 and HCT116	-	-	[111]

## 5. Conclusions and Future Perspectives

Many advancements have occurred in the production of more stable, efficient, and safe polymeric carriers, as well as formulations, and manufacturing processes will undoubtedly evolve further. Although there are few effective anticancer drugs on the pharmaceutical market, biodegradable polymeric systems have been shown to be applicable in the treatment of major diseases of global economic importance.

Cancer is still one of the leading causes of death around the world. Despite significant efforts in oncology and advancements in individualized therapy methods, significant morbidity and mortality persist. In some cancer types, conventional chemotherapy is a common treatment option. CPT belongs to the class of anticancer drugs known as Top1 inhibitors. Although this drug has potent anticancer activity, it is not used in clinical practice due to its high hydrophobicity and poor active form stability. To address these shortcomings, synthetic CPT analogues were synthesized and characterized. Given the promising response to their application, the side effects of chemotherapy still remain a challenge. Some of these effects may be reduced by the use of modern DDSs based on biodegradable polymers.

In this review, we presented various studies that reported on different combinations of biodegradable and/or bioresorbable polyester carriers for the delivery of CPT and its analogues. These materials have numerous benefits and can be used in a variety of ways. One of these features is the ability to alter their microstructure, topology, physicochemical properties, composition etc., in order to achieve controlled drug release profile. Furthermore, using such drug carriers allows for targeted therapy, which provides patients with better outcomes, fewer side-effects, and, as a result, better adherence.

According to the research updates in this publication, biodegradable and/or bioresorbable polyesters are among the essential aspects governing cancer therapy research. As a consequence, there is potential for the effective achievement of cancer-targeted DDSs based on biodegradable and/or bioresorbable polyesters, with the intention of increasing efficiency and reducing toxicity, bringing us closer to the ultimate cancer therapy system.

Despite substantial research, application of the discussed biodegradable carriers still faces a number of challenges. Biodegradable polyester DDSs can negatively interact with biomolecules due to their large surface area; the immune system can wrongfully recognize biodegradable polyester materials; some of the degradation products of the polyester matrices may exhibit toxic properties; the solvents used during their synthesis may exhibit toxic, irritating, or allergenic properties; NPs composed of biodegradable polyesters have a size similar to some proteins and can therefore interfere with the transmission of data between cells; a tiny number of the developed CPT-DDSs are characterized by a fully controlled release profile of the anticancer drug; in some cases, the phenomenon of the drug’s burst release is observed; some methods for the synthesis of CPT–macromolecular conjugates are time-consuming and expensive. In addition, other concerns, including targeting tumoral tissue, drug payload diffusion within solid tumors, and tumor heterogeneity, should be taken into consideration. Even if the proposed polyester carriers offer hope for practical application, more study is necessary to fully demonstrate the effects of innovative DDSs based on biodegradable polyesters for the delivery of CPT and its analogues in both preclinical and clinical practice.

As a final point, we would like to mention that scientists could also consider the application of biodegradable polyester DDSs for combined tumor therapy (simultaneous delivery of multiple anticancer drugs or combining conventional chemotherapeutics with other treatment modalities), as well as delivering CPT and its analogues along with a variety of other compounds (photosensitizing agents, nucleic acids, etc.), which may all more effectively utilize the adaptability of the suggested systems and their capacity to circumvent mechanisms of multidrug resistance, thus enhancing the overall anticancer effect.

## Figures and Tables

**Figure 1 ijms-24-01053-f001:**
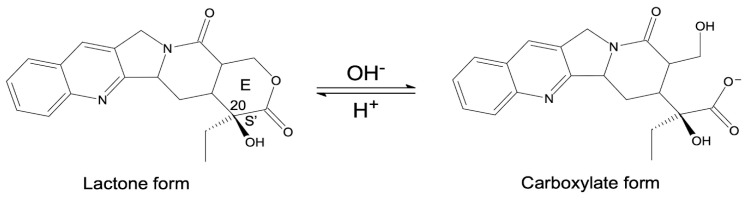
CPT structure—lactone (pharmacological active form) and carboxylate (inactive form).

**Figure 2 ijms-24-01053-f002:**
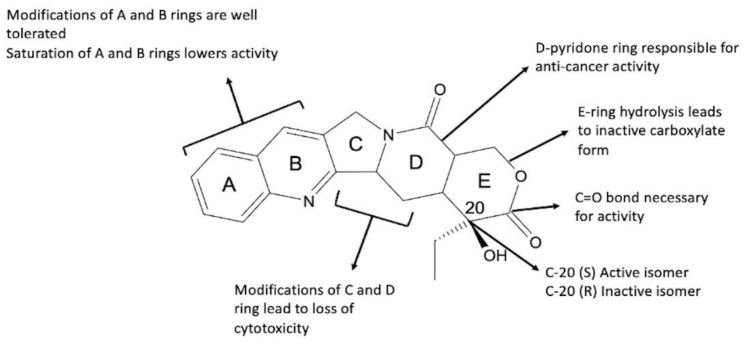
Structure activity relationship (SAR) of CPT.

**Figure 3 ijms-24-01053-f003:**
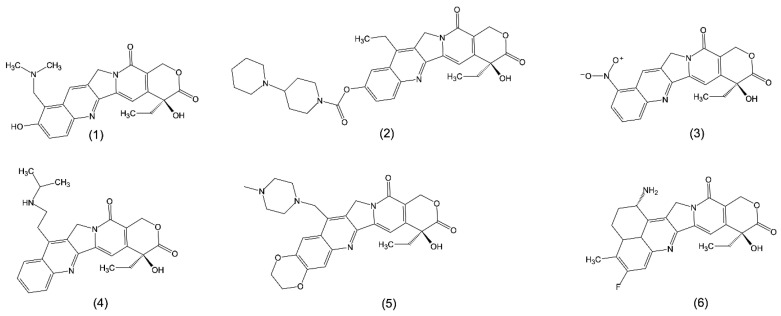
Structures of CPT analogues: topotecan (**1**); irinotecan, CPT-11 (**2**); rubitecan, 9-CPT (**3**); belotecan, CKD-602 (**4**); lurtotecan, GG-211 (**5**); exatecan, DX-8951f (**6**).

**Figure 4 ijms-24-01053-f004:**
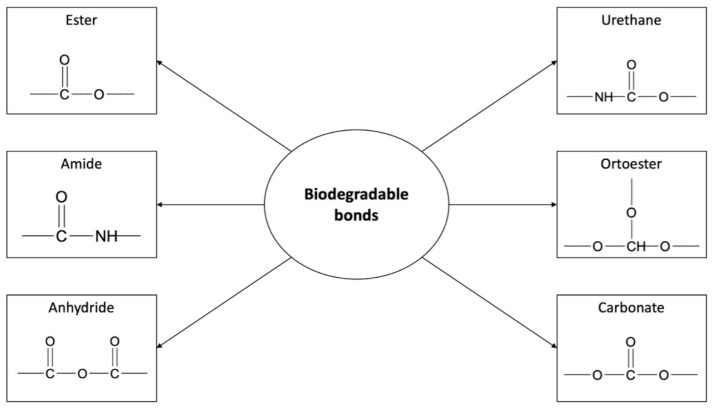
Examples of biodegradable bonds used to conjugate the drug to the polymer chain.

**Figure 5 ijms-24-01053-f005:**
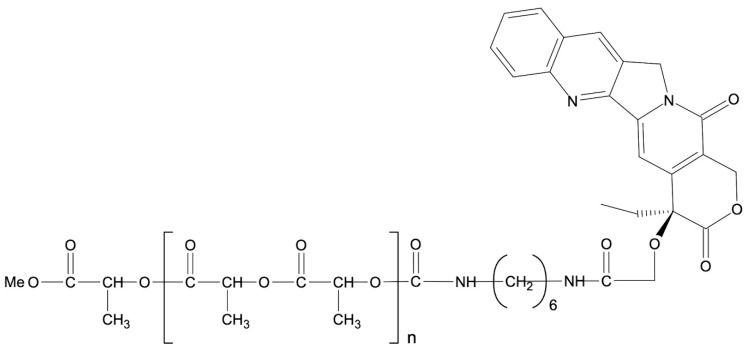
CPT–PLA conjugate obtained by coupling via 1,6-diisocyanatohexane (HMDI).

**Figure 6 ijms-24-01053-f006:**
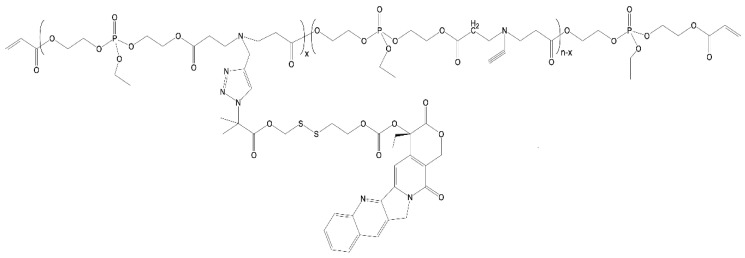
Structure of P(EAEP–PPA)–*gss*–CPT prodrug.

**Figure 7 ijms-24-01053-f007:**
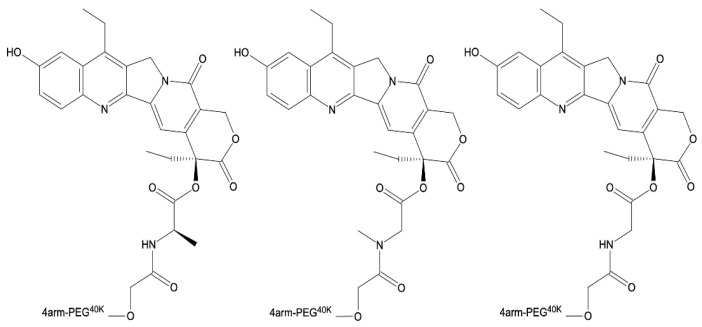
The structures of the conjugates of SN-38 and PEG-4000.

**Table 1 ijms-24-01053-t001:** Characteristics of CPT and its analogues; water solubility, drug concentration required to inhibit 50% of cell growth (IC_50_), maximum tolerated dose (MTD), and clinical trial status.

Drug	Solubility in Water	${\mathrm{IC}}_{50}\;(\mu M)$	MTD (mg/m^2^/d)	Clinical Trial Status	Reference
CPT	2.9 mM	0.046	-	-	[29]
Topotecan	100 mM	0.1	1.5	Approved	[29,40]
Irinotecan (CPT-11)	25 mM	1.14	290–320	Approved	[40,41]
Rubitecan (9-CPT)	239 μg/mL	0.085	1.5	Phase III	[29,42,43]
Belotecan (CKD-602)	77.9 μg/mL	0.094	0.5	Phase II	[43,44,45]
Lurtotecan (GG-211)	713 μg/mL	0.006	1.2	Phase II	[43,46,47]
Exatecan (DX-8951f)	221 μg/mL	0.008	0.3 for heavily pretreated patients 0.5 for minimally pretreated patients	Phase III	[43,46,47]
SN-38	11–38 μg/mL	0.09	-	-	[42,48,49,50]

## Data Availability

Not applicable.

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
