# Peer review of "The Advancement of Biodegradable Polyesters as Delivery Systems for Camptothecin and Its Analogues—A Status Report"

_ijms, 2023, doi:10.3390/ijms24021053_

Round 1

Reviewer 1 Report

The manuscript focuses on strategies to increase the bioactivity and target specificity of CTP and its analogs using polyester as carriers in various in vivo and in vitro studies. A comparison of natural and synthetic biodegradable polymers has also been presented and discussed. There are a few statements (like line 373) that need minor corrections to convey the correct message to readers. Apart from that, the manuscript is comprehensive and covers several significant aspects of this topic

Author Response

Dear Reviewer,

Please find attached our responses to your comments.

Reviewer 2 Report

This articles present interesting topic in biodegradable polymeric drug delivery systems which highlight and give useful summary of the recent approaches in the Camptothecin DDS. The author have includes some reliable and credible references, to explain the applicability of each approach, the example of the system and its result. However the authors prospective and opinion needs to be included in this review. I recommend minor revision of this article.

Author Response

(The authors gave the same response as above.)

Reviewer 3 Report

In this review, the author presented various studies that reported on different combinations of biodegradable and/or bioresorbable polyester carriers for the delivery of CPT and it analogs. The work has certain guiding significance for further developing new drug delivery systems with better performance, but some details still need to be strengthened. I think it could be published after major revisions.

1.     In the Abstract and Introduction, the authors said that the review focused on biodegradable polymeric drug delivery systems. However, the manuscript is much closer to illustrating the drugs and drug delivery applications. It should provide a section to summarize biodegradable polyester carriers and full discussion.

2.     Table 3 needs to summarize more works about advanced NPs of CPT and its analogues obtained by covalent drug conjugation.

3.     More bonding strategies of drugs with NPs should be discussed, like more non-covalent bonds, including hydrogen bonding, pi-pi stacking, halogen bonds, etc. They could also be used for drug loading, maybe should be discussed in the manuscript.

4.     Besides biocompatibility, are there any other special advantages of polymer-based nanoparticles?

5.     More advanced drug delivery works should be included in the paper, like doi.org/10.1016/j.mattod.2020.02.001, etc

6.     English still needs to be polished.

Author Response

(The authors gave the same response as above.)

Round 2

Reviewer 3 Report

The author made good revisions, no comments anymore.